# Soil Contamination in Areas Impacted by Military Activities: A Critical Review

**Parya Broomandi** [1,2,3]**, Mert Guney** [1,2,*] **, Jong Ryeol Kim** [1] **and Ferhat Karaca** [1,2]

[1]    Department of Civil and Environmental Engineering, Nazarbayev University,
       Nur-Sultan 010000, Kazakhstan; paryabroomandi@gmail.com (P.B.); jong.kim@nu.edu.kz (J.R.K.);
       ferhat.karaca@nu.edu.kz (F.K.)
[2]    The Environment and Resource Efficiency Cluster (EREC), Nazarbayev University,
       Nur-Sultan 010000, Kazakhstan
[3]    Department of Chemical Engineering, Masjed-Soleiman Branch, Islamic Azad University,
       Masjed-Soleiman 1584743311, Iran
[*]    Correspondence: mert.guney@nu.edu.kz; Tel.: +7-7172-704553; Fax: +7-7172-706054

**Abstract:** Military activities drastically affect soil properties mainly via physical/chemical disturbances during military training and warfare. The present paper aims to review (1) physical/chemical disturbances in soils following military activities, (2) approaches to characterization of contaminated military-impacted sites, and (3) advances in human health risk assessment for evaluating potential adverse impacts. A literature search mainly covering the period 2010–2020 but also including relevant selected papers published before 2010 was conducted. Selected studies (more than 160) were grouped as follows and then reviewed: ~40 on the presence of potentially toxic elements (PTEs), ~20 on energetic compounds (ECs) and chemical warfare agents (CWAs), ~40 on human health risk assessment, and generic limits/legislation, and ~60 supporting studies. Soil physical disturbances (e.g., compaction by military traffic) may drastically affect soil properties (e.g., hydraulic conductivity) causing environmental issues (e.g., increased erosion). Chemical disturbances are caused by the introduction of numerous PTEs, ECs, and CWAs and are of a wide nature. Available generic limits/legislation for these substances is limited, and their contents do not always overlap. Among numerous PTEs in military-impacted zones, Pb seems particularly problematic due to its high toxicity, abundance, and persistence. For ECs and CWAs, their highly variable physiochemical properties and biodegradability govern their specific distribution, environmental fate, and transport. Most site characterization includes proper spatial/vertical profiling, albeit without adequate consideration of contaminant speciation/fractionation. Human health risk assessment studies generally follow an agreed upon framework; however, the depth/adequacy of their use varies. Generic limits/legislation limited to a few countries do not always include all contaminants of concern, their content doesn't overlap, and scientific basis is not always clear. Thus, a comprehensive scientific framework covering a range of contaminants is needed. Overall, contaminant speciation, fractionation, and mobility have not been fully considered in numerous studies. Chemical speciation and bioaccessibility, which directly affect the results for risk characterization, should be properly integrated into risk assessment processes for accurate results.

**Keywords:** chemical warfare agents; energetic compounds; human health risk assessment; potentially toxic elements; site characterization; soil pollution

## 1. Introduction

Chemicals (particularly non-biodegradable elements and compounds) used in military ammunition and explosives may contaminate soil and surface waters, which may later cause detrimental

impacts on human health and large ecosystems around the world [1,2]. As a result, war zones with intense conflict, military training areas, shooting sports zones, and explosives and ammunition manufacturing/disposal locations are considered among the major sources of contamination for terrestrial ecosystems [3–5]. Such examples of contamination include a large list of organic and inorganic substances in soil and water, which could pose significant risks to human health as well as to the environment. For example, following introduction to the environment, the majority of potentially toxic elements (PTEs) in ammunition oxidize when exposed to air, especially in humid environments. Due to their enhanced solubility under certain chemical environments, they may become mobile/available. Potential human exposure may lead to adverse effects including damage to vital organs such as the liver and kidneys, pathology of red blood cells, and irritation of epithelial tissues [1,3,6]. For decades, extended areas belonging to military facilities have remained widely contaminated with toxic compounds, mainly explosives and munitions (and their residues) containing harmful substances including but not limited to antimony (Sb), lead (Pb), uranium (U), 2, 4-dinitrotoluene (DNT), 2, 4, 6-trinitrotoluene (TNT), 1, 3, 5-trinitro-1, 3, 5-triazacyclohexane (RDX) [7–10]. A large majority of these compounds are resistant to biological degradation or treatment and thus remain in the biosphere, becoming a contamination source potentially harmful to human health and the environment due to their possible toxic impacts [11,12].

The present study aims to review the scattered literature on the site contamination in areas impacted by military activities, focusing on (1) physical and chemical disturbances in soil induced by military activities, (2) the fate of military remains following their introduction to soil and subsequent site characterization, and (3) exposure assessment and human health risk characterization for these sites. In this context, the contaminants included in the review belong to these categories: PTEs, energetic compounds (ECs: compounds of explosives and propellants), and chemical warfare agents (CWAs).

*Search Methodology and Results*

A literature search mainly covering the period between 2010 and 2020 (but also including relevant selected papers published before this period) was conducted to find out studies on military activity-induced physical and chemical disturbances to the soil, the fate of military remains in soil and site characterization, and human health risk characterization (Google Scholar, Web of Science, Scopus). The studies on remediation techniques and the removal of military remains from soil were excluded from the present review unless they were directly linked to the aims of the present review. The search mainly focused on studies conducted on military-impacted areas, i.e., war-impacted zones, active or abandoned military bases, and detonation facilities. Multiple search inquiries were performed using combinations of key words including: "*munition", *munition site", "buried *munition", "chemical warfare", "chemical warfare agent*", "chemical weapon*", "environment* contam*", "environment* pollut*", "explosive contam*", "explosive pollut*", "exposure assess*", "geostatistic*", "heavy metal*", "metallic fragment*", "military activit*", "military area", "military conflict", "military contam*", "military explosives", "military impact", "military pollut*", "military propellant", "military zone", "risk assess*", "potentially toxic element*", "review", "risk characteriz*", "shooting range", "site assess*", "site characteriz*", "site contam*", "site investigat*", "site pollut*", "soil contam*", "soil pollut*", "soil quality", "spatial variability", trace metal*", "war activit*", "war impact* zone", "war remains", "warfare agent*", "weapon destruction facility", "world war". To the greatest possible extent, high-quality, relevant studies published in and after 2010 in scientific journals (classified as Q1 or Q2 journals by SJR (Scimago Journal & Country Rank) database) were given weight in the present literature review. The following exceptions were made on a case by case basis: (a) relevant/key studies published before 2010, (b) highly relevant studies published in journals classified as Q3 or below, (c) pertinent scientific reports, books, and legal documentation including standards regardless of their publication date. The literature search yielded more than 160 references to be considered in the present review; including around 40 studies on the presence of

PTEs and their fate/characterization in soil, around 20 on ECs and CWAs and their fate/characterization in soil, around 20 on human health risk assessment, and around 20 including generic limits and legislation regarding the protection of soils and the rehabilitation of contaminated sites by military activities. The remaining references (around 60) were supporting studies including relevant information related to the fate of PTEs, ECs, and CWAs in soil. The majority of the studies focusing on PTEs, ECs, and CWAs on contaminated sites have been conducted in Europe (Norway, Switzerland, Spain, France, Belgium, Czech Republic, Bosnia Herzegovina, Croatia, and Poland) whereas the rest were from North America (U.S., Canada), Australia, Eastern Asia (Korea), and the Middle East (Iran).

## 2. Generic Limits and Legislation for Protection of Soils and Rehabilitation of Sites Contaminated by Military Activities

Similar to industrial activities causing site contamination, military-impacted areas are also likely to pollute soil and surface waters mainly by PTEs and ECs which are also capable of leaching to the groundwater. Following military activities, toxic substances introduced to terrestrial and aquatic environments could seriously impact various organisms (even in trace amounts for some contaminants). Military training ranges and war-impacted zones may also affect economic activities of inhabitants in nearby areas including farming, breeding livestock, and fishing; consequently, the accumulation of toxic substances in soils could be a significant concern affecting their production due to potential negative impacts on crop growth, food quality, and overall environmental health [4,13,14]. Therefore, some generic limits and legislation exist aimed at the protection and rehabilitation of industrial sites, with variations from country to country due to differences in primary targets for protection and remediation; correspondingly, these mostly refer to war-impacted zones in the literature.

The soil concentrations of contaminants in military zones and war-impacted areas in literature have been typically compared to various types of defined values: (1) background soil concentrations [13,15–20], (2) agricultural soil concentrations [9,17,21,22], (3) generic industrial, urban, and/or recreational soil concentrations [9,13,16,23–29], and (4) military range soil concentrations. Some reference values, as well as limits for PTE and CM concentrations in soils, are summarized in Tables 1 and 2, respectively. To the authors' knowledge, although some preliminary guidelines were calculated by the Biotechnology Research Institute for some explosives in soils [30,31]; the only official guidelines for explosive materials in soils come from the U.S. EPA [31]. This approach is similar to the one used for the control of industrial contamination, based on enforced maximum allowable concentrations of substances in the environment. However, the availability of these generic limits and legislation is limited to a few countries. Furthermore, these limits do not always overlap in terms of content, and their scientific basis is not always clear. Finally, although these limits would be imposed in peacetime for formerly contaminated sites or training grounds, their usability is highly questionable under ongoing conflict scenarios.

**Table 1.** Reference/limit values (mg kg$^{-1}$) used for evaluating concentrations of potentially toxic elements (PTEs) in soil samples in reviewed studies.

| Studies | Country | Ag | As | Be | Cd | Co | Cr | Cu | Hg | Ni | Pb | Sb | V | Zn | Comments |
|---|---|---|---|---|---|---|---|---|---|---|---|---|---|---|---|
| [9] | Bosnia and Herzegovina | - * | - | - | 1 | - | - | 65 | - | 40 | 80 | - | - | 150 | Maximum allowed values for silty loam soil |
| [24] | U.S. | 4 | - | - | 0.36 | - | 26 | 28 | 0.1 [a] | 38 | 11 | - | - | 46 | U.S.EPA eco-screening levels for most sensitive receptors |
| [24] | Spain | 78 | - | - | 14 | - | 1100 [b] | 630 | 4.7 [c] | 760 | 400 | - | - | 1000 | Soil screening level for unrestricted land use on shallow soils overlying usable groundwater resources |
| [22] | Croatia [d] | - | - | - | 0–0.5 | - | 0–40 | 0–60 | 0–0.5 | 0–30 | 0–50 | - | - | 0–60 | Maximum allowed concentrations for silty loamy soils (for agricultural use) |
| [22] | Croatia [e] | - | - | - | 0.5–1 | - | 40–80 | 60–90 | 0.5–1 | 30–50 | 50–100 | - | - | 60–150 | |
| [22] | Croatia [f] | - | - | - | 1–2 | - | 80–120 | 90–120 | 1–1.5 | 50–75 | 100–150 | - | - | 150–200 | |
| [17,21] | Korea | - | - | - | 4 | - | - | 150 | - | - | 200 | 40 [g] | - | 300 | Korean regulation for agricultural soils [32] |
| [17,18] | Norway | - | - | - | - | - | - | - | - | - | 60 | 40 | - | - | Background levels [33] |
| [19] | Spain | - | - | - | - | - | 80 | 45 | - | 65 | 55 | - | - | 100 | Background levels |
| [19] | Spain | - | - | - | - | - | 100 | 100 | - | 100 | 100 | - | - | 500 | Generic reference level for urban and recreational soils [25] |
| [16,29] | Korea | - | - | - | - | - | - | 2000 | - | - | 700 | 5 | - | - | Korean regulation for shooting range soils [34] |
| [28] | Czech Republic | - | 30 | 7 | 1 | 50 | 200 | 100 | - | - | 140 | - | - | 200 | Maximum permissible limits [35] |
| [26,27] | Canada | - | - | - | - | - | - | 100 [h] | - | - | 500 [h] | - | - | 500 [h] | Threshold values for commercial or industrial use & for residential use [36–38] |
| [26,27] | Canada | - | - | - | - | - | - | 63 [i] | - | - | 140 [i] | 20 [i] | - | - | |
| [26,27] | Canada | - | - | - | - | - | - | 500 [h] | - | - | 1000 [h] | - | - | 1500 [h] | |
| [26,27] | Canada | - | - | - | - | - | - | 91 [i] | - | - | 600 [i] | 40 [i] | - | - | |

**Table 1.** *Cont.*

| Studies | Country | Ag | As | Be | Cd | Co | Cr | Cu | Hg | Ni | Pb | Sb | V | Zn | Comments |
|---|---|---|---|---|---|---|---|---|---|---|---|---|---|---|---|
| [16,20] | Belgium | - | 16 | - | - | - | - | 20 | 0.1 | 16 | 31 | - | - | 77 | Background values for standard agricultural soils defined as containing 10% clay and 2% organic matter [39] |
| [23] | France | - | 125 | - | - | - | 1000 | - | - | 350 | 1000 | - | - | - | - |
| | France | - | 55 | - | - | 240 | 380 | 190 | - | 210 | 530 | - | - | 720 | Soil intervention values [40] |
| [15] | Switzerland | - | - | - | - | - | - | 9.4 | - | - | - | - | - | - | Background levels |

* -: Not available; [a] Inorganic Hg; [b] Total Cr; [c] Total Hg; [d] Sandy soil; [e] Silty-loam soil; [f] Sandy-clay soil; [g] Norwegian soil quality guideline for Sb; [h] Criterion C for commercial or industrial use; [i] Criterion B for residential use.

**Table 2.** Reference/limit values (mg kg$^{-1}$) used for evaluating concentrations of energetic compounds (ECs) in in soil samples in reviewed studies.

| Resource | Country | HMX | RDX | TNT | 4ADNT | NG | 2,4-DNT | 2,6-DNT | 1,3-DNB | 2,4,6-TNT | Comment |
|---|---|---|---|---|---|---|---|---|---|---|---|
| [41] | U.S. | 51,000 | 26 | 95 | - * | 200 | 2000 | 1000 | - | - | Risk-based concentrations in soil (industrial) by U.S.EPA Region 3 [41] |
| | U.S. | 3900 | 5.8 | 21 | - | 46 | 160 | 78 | - | - | Risk-based concentrations in soil (residential) by U.S.EPA Region 3 [41] |
| [13] | Canada | 32 | 4.7 | 3.7 | - | 65 | 11 | 8.5 | - | - | Preliminary soil quality guideline for the environment [30] |
| | Canada | 4100 | 250 | 41 | - | 2500 | 0.14 | 0.14 | - | - | Preliminary soil quality guideline for human health [30] |
| | Canada | 13 | 7.6 | 31 | - | 2.4 | 130 | 130 | - | - | Preliminary soil quality guideline to protect aquatic life in case of groundwater discharge [30] |
| [23] | France | - | - | - | 100 | - | 100 | - | 75 | 8 | German soil investigation values proposed for parks recreational areas [42] |

* -: Not available.

## 3. Physical Impact of Military Activities on Soil

The morphology of military- impacted soils exhibits anthropogenic changes to its features. The disturbance or removal of soil material for avoiding or limiting offensive attempts as well as developing escape plans are some examples of activities which alter its physical characteristics, including the hydrological behavior in the area. The constructed underground cities or tunnels are impressive examples of excavation for defensive and escape plans, which causes perturbation or removal of vast tons of lateritic soil. Warfare tunneling was widely practiced during the American Civil War, World War II, and the Vietnam War [43–46]. Another example is bombs and shells, causing cratering during their use in conflicts or troop exercises. Explosions are capable of removing large quantities of earth, creating a hollow. The soil found in a hollow is compacted, perturbed, and contaminated by metallic fragments and ash. This type of soil disturbance is called bombturbation and includes the displacement of removed soil from the hollow to its proximity [11,14]. Bombturbation disrupts the landscape as it mixes soil horizons, resulting in a significant transformation in topography. Buried anti-tank and anti-personnel mines are also responsible for soil disturbance if they explode. In fact, mine emplacement itself could cause significant soil perturbation. Following its activation, the soil surrounding a mine quickly becomes contaminated by plastic and metallic fragments as well as by the residues of explosives [11].

Military traffic, which includes maneuvers of wheeled or tracked heavy vehicles, is another significant factor that may affect soil. Compaction is the main negative impact of military traffic on soil, significantly modifying the hydraulic properties of soil and also making soils more vulnerable to erosion and runoff [47–50]. In saturated soils, excess loading could cause liquefaction, resulting in various issues, including mud formation [51].

Finally, instigation of fires in crops or forests has numerous adverse consequences on the physical properties of soil [11,52–54]. The most indirect significant impact is susceptibility to erosion, which rapidly propagates on steep, incinerated surfaces. Furthermore, due to the fire-induced formation of a hydrophobic layer at shallow depths that prevents/limits water infiltration, these areas are highly vulnerable to runoff and erosion [11,55,56], which may result in shallow landslides and debris flow [57]. A cause-and-effect diagram regarding the physical impacts of military activities on the soil is presented in Figure 1.

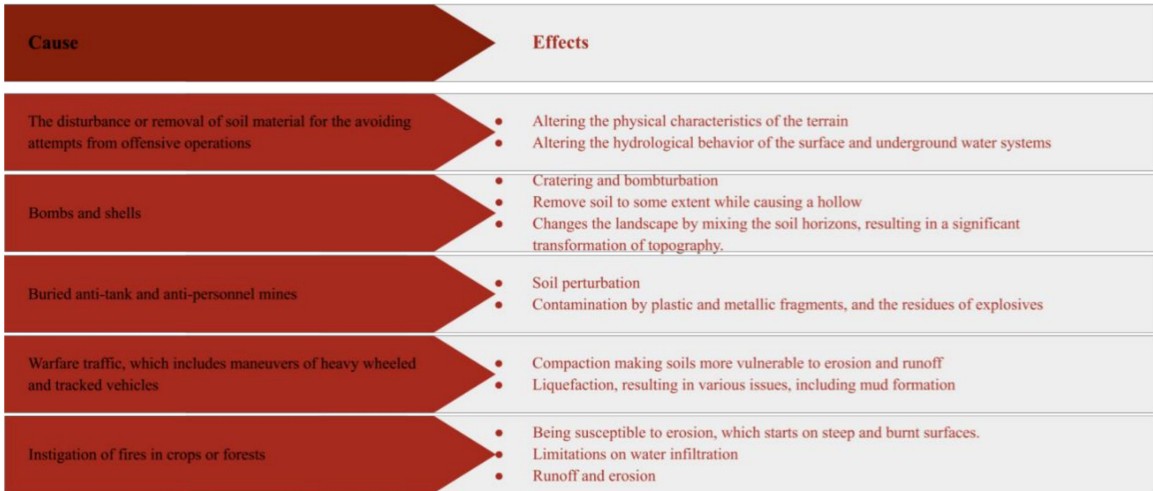

**Figure 1.** Cause-and-effect diagram for physical impacts of warfare on soil.

## 4. Chemical Impact of Military Activities on Soil

### 4.1. Soil Contamination by Potentially Toxic Elements

Metallic remains are among the longest-remaining war residuals in conflict-impacted zones. The residence time of these remains depends mainly on soil redox properties, namely the proton (pH) and electron (pe) activities. With time, PTEs may mobilize, and then new resulting minerals (predominantly oxides) could precipitate starting from supersaturated soil solution. Therefore, conflict areas around the world could represent significant sinks as well as potential sources of pollution by PTEs both in soil and water [21,58]. Bullets, for instance, can lead to a release of Pb following complex mineralogical and chemical processes in the soil some of which conclude with the precipitation of insoluble minerals or plant uptake [59]. As a major PTE in military-impacted zones, Pb partitioned in different soil fractions may be initially inert but then later become reactive following changing soil conditions (e.g., pH, moisture, OM) or when its quantities in soil exceed the soil-holding capacity [60]. Other PTEs often released to soil by weapon residues include Sb, chromium (Cr), arsenic (As), mercury (Hg), nickel (Ni), zinc (Zn), and cadmium (Cd) [21,61]. It has been recently shown that plant species across military ranges can accumulate PTEs (specifically Cr, Cu, Ni, Pb, Sb, and Zn) and that the exact relationship between a particular PTE and where it accumulates depends on the species [62].

Numerous studies have reported elevated concentrations of PTEs in soil samples collected from war-impacted areas and military training grounds [6,7,9,13,15–17,19–24,26–29,63–70] with their reported values summarized in Table 3.

In order to compare these, a soil pollution index (SPI) of the reviewed areas has been calculated using individual pollution indices ($SPI_i$) for each PTE investigated, which are individual enrichment ratios of these PTEs. The SPI value of a reviewed zone is calculated as follows,

$$SPI = \left( \prod_{i=1}^{n} \frac{C_i}{C_0} \right)^{1/n} \tag{1}$$

where $C_0$ is the measured background concentration of the selected PTE as a warfare tracer [71] and $C_i$ is its soil concentration. In other words, the geometric mean of the individual indices for PTEs reported in each case provides the SPI value for the study reviewed. Only the results for PTEs being common warfare metal tracers (e.g., Cu, Pb, Zn, and Sb) have been included in the assessment, and SPI is limited to the extent of their availability of reporting in individual studies. The SPI levels calculated for the reviewed studies are presented in Figure 2. The sites were then classified using the following scale: (1) extremely contaminated (SPI > 10,000), (2) very highly contaminated (1000 < SPI < 10,000), (3) highly contaminated (100 < SPI < 1000), (4) moderately contaminated (10 < SPI < 100), and (5) marginally contaminated (SPI < 10).

**Table 3.** Concentrations of common PTEs (mg kg$^{-1}$) in military-impacted soils in reviewed studies conducted on military base areas (MBA) and war-impacted areas (WIA).

| Reference | Site | Activity | Soil pH | Ag | As | Ba | Cd | Co | Cr | Cu | Hg | Mn | Ni | Pb | Sb | Ti | V | Zn | Zr |
|---|---|---|---|---|---|---|---|---|---|---|---|---|---|---|---|---|---|---|---|
| [9] | Bosnia Herzegovina | MBA | 9.2–9.9 | - | - | - | 0.8–6.1 | - | - | 23.6–443 | - | - | 40.4–73.6 | 27.7–40.9 | - | - | - | 91.7–238 | - |
| [69] | Switzerland | MBA | 6.1–8.2 | - | - | - | - | - | - | 63–66 | - | 480–860 | 55–61 | 500–620 | 17–21 | - | - | 100–110 | - |
| [21] | Korea | MBA | 6–6.8 | - | - | - | 7.45–8.11 | - | - | 318–562 | - | - | -* | 3918–18,609 | 26–108 | - | - | 104–123 | - |
| [17] | Norway | MBA | 4.8–6.5 | - | - | - | - | - | - | 41–88 | - | - | - | 356–1112 | 40–123 | - | - | - | - |
| [19] | Spain | MBA | 3.72–6.75 | - | - | - | - | - | 40–79 | 19–98 | - | - | 11–33 | 55–6309 | - | - | - | 34–264 | - |
| [29] | Korea | MBA | 8 | - | - | - | - | - | - | 1168 | - | - | - | 17,468 | 164 | - | - | - | - |
| [28] | Czech Republic | MBA | 5.6–7.7 | - | 5.33 | - | 0.235 | 3.81 | 18.4 | 6.91 | - | - | 10.7 | 15.5 | 3.33 | - | - | 34.3 | - |
| [27] | Canada | MBA | - | - | - | - | - | - | - | 245 | - | - | - | 3368 | 73 | - | - | 177 | - |
| [65] | U.S. | MBA | 4.9 | - | 2.47–2.67 | - | - | - | 36.27–38.4 | 65.67–118.77 | - | 91.57–107.33 | 33.777–57.33 | 17.85–19.30 | 0.08–0.12 | - | - | 54.8–58.27 | - |
| [26] | Canada | MBA | 5.9–8.1 | - | - | - | - | - | - | 1760 | - | - | - | 43,300 | 780 | - | - | 355 | - |
| [68] | Australia | MBA | 5.3–6.4 | - | 0.25–9.55 | - | - | - | - | 0.43–1250 | - | - | 0.48–8.97 | 1.18–10,403 | 1–252 | - | - | 0.99–179 | - |
| [67] | U.S. | MBA | 6.11–6.72 | - | - | - | - | - | - | - | - | - | - | 10,068–70,350 | - | - | - | - | - |
| [66] | Canada | MBA | - | - | - | - | - | - | - | 1830–7720 | - | - | - | 14,400–27,100 | 150–570 | - | - | 260–1080 | - |
| [63] | U.S. | MBA | 4.4–8.19 | - | 2.8–27.9 | - | - | - | - | 223–2936 | - | 83–930 | 3–33 | 4549–24,484 | 7–91 | - | - | 102–284 | - |
| [13] | Canada | MBA | - | - | 0.8–10 | 24.2–75 | 0.1–15.2 | - | 4–24.1 | 2.5–154 | - | - | 3–21 | 5–53.8 | - | - | 6.9–35.5 | 11.9–120 | - |
| [6] | Korea | MBA | - | - | - | - | 0.0735–0.22 | - | - | 3.12–83 | - | - | - | 3.48–16.9 | - | - | - | - | - |
| [15] | Switzerland | MBA | 3.2–3.6 | - | - | - | - | - | - | 32–552.3 | - | - | 21.3–114.7 | 429–80,935 | 6.2–4022.4 | - | - | 60.3–128.3 | - |
| [64] | Iran (Iran-Iraq War, 1980–1988) | WIA | 8 | - | 3.9 | 93 | - | 13 | 156 | 40 | 2.25 | 443 | 110 | 36 | 8 | 3000 | 63 | 4420 | 100 |
| [24] | Spain (WWII, 1939–1945) | WIA | - | 1.4–42 | - | - | 15–23 | - | 60–115 | 1403–11,860 | 0.142–0.624 | 960–2492 | 19–96 | 1555–2000 | - | - | - | 2805–9019 | - |
| [7] | Poland (WWI, 1914–1918; WWII, 1939–1945) | WIA | 5.3–5.9 | - | - | - | - | - | - | - | 0.4162 | - | - | - | - | - | - | - | - |
| [22] | Croatia, (War of Independence, 1991–1995) | WIA | 4.8–7.2 | - | - | - | 0.13 | - | 32 | 13 | 0.07 | 506 | 19 | 17 | - | - | - | 53 | - |
| [70] | France (WWI, 1914–1918) | WIA | 5.3–5.9 | - | 1937–72,820 | - | - | - | - | 1451–9113 | - | - | - | 968–5777 | - | - | - | 10,660–90,190 | - |
| [16] | Belgium, (WWI, 1914–1918) | WIA | - | - | - | - | - | - | - | 23.3 | - | - | - | 47.6 | - | - | - | - | - |
| [20] | Belgium (WWI, 1914–1918) | WIA | - | - | - | - | - | - | - | 26.9 | - | - | - | - | - | - | - | - | - |
| [23] | France (WWI, 1914–1918) | WIA | 4.4–5.8 | - | 59–136,770 | - | - | 5–7 | 22–39 | 20–15,755 | - | 99–840 | 8–17 | 766–26,398 | - | - | - | 399–133,237 | - |

* -: Not available.

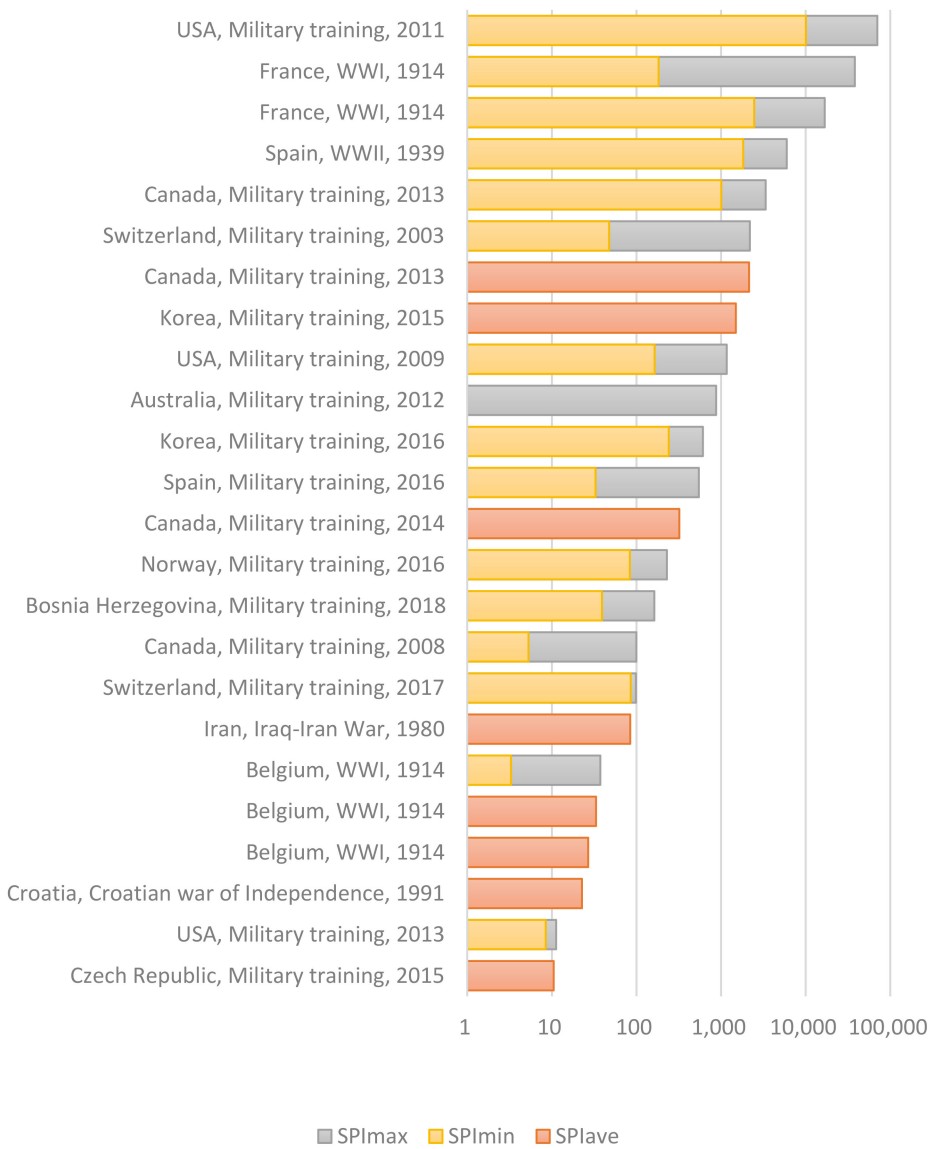

**Figure 2.** Soil Pollution Index (SPI); maximum, minimum, and average values calculated for reviewed studies.

Overall, the critical review indicated that the contamination distribution was typically dependent on firing activity, soil properties, soil exposure time, and climate, which expectedly show significant variations among the reviewed sites. The first group of cases included extremely contaminated sites (SPI > 10,000): a U.S. military training area in Florida [67] and World War I (WWI) impacted zones in France [23,70]. The soil samples from Florida, U.S. were mostly sandy, having acidic-neutral pH (from 6.11 to 6.72), low organic matter (OM) content (0.21–1.01% wt.), and moderate cation exchange capacity (CEC: from 8.34 cmol kg$^{-1}$ to 24.8 cmol kg$^{-1}$). Samples had the most Pb accumulation (60–70%) in very coarse sand fraction (1.0–2.0 mm), with concentrations being very high and ranging from 10,068 mg kg$^{-1}$ to 70,350 mg kg$^{-1}$ [67]. Furthermore, this study reported only Pb levels, i.e., the assessment does not reflect the entire profile of PTE contamination. The soil samples from Verdun, France had slightly acidic pH (as low as 5.3), and the locations with no vegetation cover were among the most contaminated, whereas the samples from areas covered by forest vegetation and hummus were the least contaminated. These WWI zones have large amounts of ammunition and shells stored at the end of the war. All soil samples contained fine sand (except one, clay loam). As, Zn, Cu, and Pb

were dominantly found in coarse and fine sand fractions, while K, Al, and Si were mainly present in loam and clay content [23,70]. Particularly, As concentrations were extremely high. As could also be abundant in a wide variety of soils due to geogenic sources [72], unlike most PTEs studied in the present review. However, the extremely high concentrations reported by Bausinger et al. [23] and Thouin et al. [70] were also well beyond the ranges in soils containing geogenic As. Furthermore, the presence of As has also been linked to military activities by the authors [23,70].

The very highly contaminated military-impacted zones (SPI > 1000) were military training zones from Canada [26,66], Korea [29], and the U.S. [63] along with a World War II (WWII) impacted area from Spain [24]. The soil samples from the Canadian sites were taken from a bullet backstop berm in a small arms shooting range [26] and a military base [66]. Both cases indicated highly elevated Pb and Sb, along with Cu and Zn elevated to a lesser extent. In the Swiss study, the surface samples (0–10 cm) were collected from an army shooting range exhibiting an acidic pH (3.6), moderate CEC value (119 mmol kg$^{-1}$) and organic carbon (OC) (1.48% wt.) with a sandy loam texture [15] and showed similar contamination profiles to the Canadian sites. The soil samples from military shooting ranges in Korea (sandy loam, pH: 8, CEC: 2.72 cmol kg$^{-1}$) had Pb concentrations exceeding the Korean regulatory warning levels for shooting range soils by 25 times. The Cu and Sb contents did not exceed the regulatory levels for military sites (2000 mg kg$^{-1}$) but were above the standard warning level for Sb (>5 mg kg$^{-1}$) and Cu (>150 mg kg$^{-1}$) for agricultural soil [29]. Samples collected in eight different states in the U.S. had variable pH values ranging from extremely acidic (4.4) to moderately alkaline (pH 8.2). CEC also widely varied (from 0.95 to 28.6 meq 100 g$^{-1}$), corresponding well with the general soil textural classifications (i.e., low values associated with sandy soils, higher values with clay-rich soils). Pb and Cu exhibited high concentrations whereas the concentrations of Sb, As, Ni, and Zn were much lower [63]. Surface soil samples from dumpsites located in Spain dating back to the immediate post-WWII clean-up period were collected within an approximate 3 m radius at each site. Three out of 35 of the Spanish sites showed significant enrichment profiles by all PTEs analyzed, particularly by Zn, Pb, Hg, Cu, Cd, and Ag.

The highly contaminated military-impacted areas (SPI >100) were military training areas from Australia [68], Korea [21], Spain [19], Canada [27], Norway [17], and Bosnia Herzegovina [9]. For the site in Australia, soil sampling was conducted at four shooting ranges. The soil texture was generally sandy, with small amounts of silt and clay, having samples with varying pH from acidic (5.3) alkaline (9.3). CEC values also varied from 1.8 to 33.2 cmol kg$^{-1}$ and OC content was low (0.06–0.76% wt.) [68]. The highest contamination was observed for Pb and Sb, with much lower concentrations for As, Cu, and Zn [68]. In Korea, the most abundant PTEs in two shooting ranges in order were Pb > Cd = Sb >> Cu [21]. Compared to Korean warning levels, the total average PTE concentrations were 20 to 93 times higher for Pb, two–four times higher for Cu, and 2 times higher for Cd (no regulatory limit for Sb). In Spain, surface soil samples collected from a small shooting range had a different texture, i.e., from fine loamy sand to sandy loam, which was attributed to the movement and transportation of soil particles between different zones of shooting range (with scarce vegetation) by on-ground activities [19]. The soil pH varied from strongly acidic (3.72) to neutral (pH = 6.75). The OM content was very high (between 4.7 and 17.31% wt.), and effective CEC values were from 4.53 to 9.61 cmol kg$^{-1}$. The highest contamination was for Pb, followed by Zn (<10) mainly found in sandy loam soils; whereas the concentrations of Ni, Cr, and Cu were below or very close to background values. The Canadian military base study by [27] is a different investigation in the same base [26] yielding lower but still elevated PTE concentrations. Soil samples from shooting ranges in Norway were studied for their total concentrations of Sb and Pb. Soil samples were generally classified acidic with pH values ranging from 5.3–6 and had total OM of 0.41% wt. [17]. The findings were in agreement with the concentration ranges previously reported for shooting ranges [73–75]. It is notable that there was a relatively constant ratio of Sb:Pb concentration (0.110 ± 0.001), which seems to reflect the Sb:Pb ratio in the core of the original ammunition mainly used at this location [15,17]. In Bosnia and Herzegovina, soil samples were taken from an open detonation pit from different depths [9].

Generally, soils from observed locations could be classified as a silty loam soil having alkaline pH (9.2–9.9). In this case, Pb was less elevated than the other cases in the same contamination category.

The moderately contaminated military-impacted zones (SPI > 10) were from Canada [13], Switzerland [69], Iran [64], Korea [6], Belgium [16,20], Croatia [22], the U.S. [65], and the Czech Republic [28]. In Canada, soil samples were collected from four training ranges, and their PTE (Cd, Cu, Pb, Zn, Ni, Cr, As, V, and Ba) pollution levels showed significant variation [13]. The most polluted locations were attributed to the frequency of munitions use (i.e., rockets and dummy bombs) where pollution levels of PTEs were 24 times the background levels, whereas these levels did not exceed more than five times the background levels in other ranges. In Switzerland, two soil samples from shooting ranges were investigated for their content of PTEs (Sb, Pb, Cu, Ni, Zn, Mn, and Fe) [69]. The abundance of PTEs was in the order of Pb > Zn = Cu. Soil samples from the Shalamcheh war zone in Iran were studied for PTE contamination (Pb, Hg, Co, Cr, Zn, As, Fe, Sb, Ni, and Cu) [64]. General soil characterizations showed that soil samples were mostly sand with pH values of 8, and OM content of 0.061% wt. Br, Cl, Mo, S, Zn, and Hg have been detected as slightly elevated PTEs with anthropogenic origins, mainly from Iran-Iraq War (1980–1988) remains. Furthermore, detected elevated concentrations of Cl and S in this area could be the result of using CWAs (mustard gas) in the Iran-Iraq War [64,76]. Topsoil samples collected in the North Prasnik area in Croatia were studied for PTEs (Cd, Cu, Hg, Pb, Cr, Ni, Zn, and Mn) [22]. The prevailing soil texture was classified as silty loamy and had pH value ranging from 4.8 to 7.2. CEC ranged from 13 to 29 meq 100 g$^{-1}$. According to maximum allowable concentrations (MACs) of soil for agricultural use in Croatia, the measured Cd, Cu, Hg, Cr, Ni, Zn, and Pb levels were low for dominant soil type (silty loamy soils). Mn was not considered a pollutant by the Croatian legislation [77]; thus, there was no proposed MAC. For agricultural soils, the MAC value for Mn had an amount ranging from 1500 to 3000 mg kg$^{-1}$, which was higher than the maximum Mn concentration on the site [22]. A former military training site in the Czech Republic has been studied for PTEs (As, Cd, Co, Cr, Ni, Cu, Pb, and Zn) in 2015 where military activities in the area ended in 1991 [28]. The soil pH varied between 5.6 and 7.7, with the most abundant being slightly alkaline varying between 7.1 and 7.7 [28]. CEC levels varied between 74 and 264 mmol kg$^{-1}$. The analyses of soil samples have shown that measured values of the average pseudo-total concentration of As, Cd, Co, Cr, Ni, Cu, Pb, and Zn were not elevated [28]. Soil samples from Camp Edwards military range in the U.S. were studied for concentrations of Pb, Cr, Cu, Ni, Zn, Mn, Fe, Mo, V, Ca, and As [65]. General soil characterizations showed that soil samples were mostly sand with pH values around 4.9. The topsoil examined around Ypres (West-Flanders, Belgium) in an area of 3144 km$^2$, is a battlefield in WWI, where millions of copper (Cu)—containing shells were fired [20]. Since there is no evidence of other anthropogenic sources, e.g., industrial contamination, it could be concluded that around 2800 tons of Cu were supplied to the upper soil (0.5 m) by the war activities. Also, in the same area based on 731 data points, it was witnessed that the spatial patterns of Cr and Ni were related to soil texture variations, but those of Pb and Cu were mainly related to war activities [16]. Soil samples from a military gunnery range in Korea were studied for their concentrations of Cd, Cu, and Pb [6]. The concentrations of Pb and Cu on the site were relatively high, especially for Cu in a particular zone due to the extensive usage of bombing, exceeding the Korean Environmental Standards.

Overall, in several cases, the PTE from military activities was mainly Pb and its co-contaminants including Sb, Cu, Zn, Ni, and As. The contamination was not solely limited to firing ranges, and it could be transported by accidental ammunition disposal or drainage runoff. Recently, Pb-free munitions, including steel, iron, and various alloys have become available, but they can still be a possible source of contamination by other PTEs such as Bi, Cr, and Ni. These more recently introduced munitions could cause environmental problems due to their higher corrosion rate in comparison to that of Pb [66,78–80], implying more important roles of soil properties and climate factors [19]. Soil texture has an important role in the mobility and retention of chemical elements. Generally, coarser soils with higher fractions of larger particles (e.g., sand content) have less preference for sorption whereas soils with a higher fraction of fine particles have a higher specific surface area (and thus reactive surface) as in the case of

clay minerals ad Mg and Fe—oxides/hydroxides. Therefore, soils with a dominant texture of silty loam could be characterized as favorable from the aspect of PTEs immobilization [9,81].

The pH value of soil has a great influence on the behavior of PTEs, determining their solubility and availability. An alkaline pH can, but does not always, positively affect the immobilization of PTEs whereas, in an acidic environment, metallic cations are almost always more mobile, i.e., larger quantities could be released to the soil solution and thus potentially become toxic to plants [9,82]. Slightly alkaline or neutral environments generally provide the highest PTE retention, where PTEs are immobilized, including via the formation carbonates or hydroxides.

OM is another key soil property which may be significantly affected by land use management practices [83] which also directly controls the mobility of PTEs in soil. Indeed, to increase OM and consequently CEC via soil amendments is one way to valorize soils contaminated by PTEs by achieving lower PTE overall mobility, plant bioavailability, and oral bioaccessibility [84]. It should be noted that manipulations improving the soil conditions may reduce the mobility of targeted PTEs while increasing the mobile fraction of other PTEs or impact soil biota (e.g., the use of nanomaterials for shooting range soils [85], so these approaches should be used carefully.

To summarize, it could be expected that specific characteristics of acidic soils, sandy texture, low CEC, and low OM (separately or in combination) will increase the leaching/mobilization processes governing the mobility of PTEs and thus result in higher elemental concentrations in soil solution; whereas alkaline soils, loam/silty loam/clay texture, high CEC, and high OM (separately or in combination) would limit PTE mobility and thus restrain their concentrations in soil solution and subsequent translocation into plant tissues. Overall, evaluating the PTE contamination in a military-impacted area solely based on the total concentrations of PTEs may not always be enough to draw solid conclusions, and characterization for additional parameters may significantly strengthen the confidence of contaminated site assessment.

*4.2. Contamination and Transformation of Soil by Energetic Compounds*

ECs are comprised of explosives, propellants, and fuels; they have been and still are common soil contaminants. Explosive materials are mainly nitroaromatic compounds (most frequently encountered are hexahydro-1,3,5-trinitro-1,3,5-triazine (RDX), 2,4,6-trinitrotoluene (TNT), and octahydro-1,3,5,7-tetranitro-1,3,5,7-tetrazocine (HMX); other common compounds include nitroglycerin, 1,3,5-trinitrobenzene (TNB), dinitrobenzene (DNB), 2,4,6-trinitrophenol, and N-methyl-N,2,4,6-tetranitroaniline (tetryl)). Some explosives such as nitroglycerin (NG) and pentaerythritol tetranitrate (PETN) are seldom found in soils; therefore, the studies addressing them in the soil are scarce [86,87]. Table 4 shows the physio-chemical characteristics of selected explosive compounds, each of which have the potential to pose specific environmental risks as discussed later [4,11,88–90]. Table 5 presents a summary of the sampling studies of explosive compounds for surface and subsurface soil at different military ranges [6,8,79,91–112].



**Table 4.** Physio-chemical characteristics of common ECs [89].

| Energetic Compound | CAS Number | MW (g) | Melting Point (°C) | Boiling Point (°C) | Vapor Pressure (mm Hg) | Henry's Law Constant (atm m$^3$ mol$^{-1}$) | Water Solubility (mg L$^{-1}$) | Temperature at Solubility Reported (°C) | Log P |
|---|---|---|---|---|---|---|---|---|---|
| 1,3-Dinitrobenzene (1,3-DNB) | 99-65-0 | 168.11 | 89–90 | 302.8 | $2.00 \times 10^{-4}$ | $4.90 \times 10^{-8}$ | 533 | 25 | 1.49 |
| 2,4-Dinitrophenol | 51-28-5 | 184.11 | 112–114 | sublimes | $3.90 \times 10^{-4}$ | $8.60 \times 10^{-8}$ | 2790 | 20 | 1.67 |
| 2,3-Dinitrotoluene | 602-01-7 | 182.14 | 59–61 | - * | $4.00 \times 10^{-4}$ | $9.30 \times 10^{-8}$ | - | - | 2.2 |
| 2,4-Dinitrotoluene | 121-14-2 | 182.14 | 71 | 300 | $1.47 \times 10^{-4}$ | $1.30 \times 10^{-7}$ | 270 | 22 | 1.98 |
| 2,5-Dinitrotoluene | 619-15-8 | 182.14 | 52.5 | - | $4.00 \times 10^{-4}$ | $9.30 \times 10^{-8}$ | - | - | 2.2 |
| 2,6-Dinitrotoluene | 606-20-2 | 182.14 | 66 | 285 | $5.67 \times 10^{-4}$ | $9.26 \times 10^{-8}$ | 180 | 20 | 2.1 |
| 3,4-Dinitrotoluene | 610-39-9 | 182.14 | 58.3 | - | $4.00 \times 10^{-4}$ | $9.30 \times 10^{-8}$ | 100 | 25 | 2.08 |
| 3,5-Dinitrotoluene | 618-85-9 | 182.14 | 93 | - | $1.90 \times 10^{-3}$ | $9.30 \times 10^{-8}$ | - | - | 2.28 |
| 2-Nitrophenol | 88-75-5 | 139.11 | 44–45 | 216 | 0.113 | $1.30 \times 10^{-5}$ | 2100 | 20 | 1.79 |
| 3-Nitrophenol | 554-84-7 | 139.11 | 97 | 194 | 0.1 | $2.00 \times 10^{-9}$ | 13,550 | 25 | 2 |
| 4-Nitrophenol | 100-02-7 | 139.11 | 113–114 | 279 | $9.79 \times 10^{-5}$ | $1.30 \times 10^{-8}$ | 16,000 | 25 | 1.91 |
| 1,3,5-Trinitrobenzene (1,3,5-TNB) | 99-35-4 | 213.11 | 122.5 | 315 | $3.20 \times 10^{-6}$ | $3.08 \times 10^{-9}$ | 340 | 20 | 1.1 |
| 2,4,6-Trinitrophenol | 88-89-1 | 229.1 | 122–123 °C | 300, explodes | $7.50 \times 10^{-7}$ | $1.70 \times 10^{-8}$ | 12.7 | 25 | 1.44 |
| 2,4,6-Trinitrotoluene (TNT) | 118-96-7 | 227.13 | 80.1 | 240, explodes | $1.99 \times 10^{-4}$ | $4.57 \times 10^{-7}$ | 130 | 20 | 1.6 |
| Ammonium perchlorate | 14797-73-0 | 117.49 | decomposes | decomposes | negligible | negligible | $2.49 \times 10^5$ | 20 | negligible |
| Cyclotrimethyl-enetrinitramine (RDX) | 121-82-4 | 222.26 | 205–206 | decomposes | $4.10 \times 10^{-9}$ | $6.30 \times 10^{-8}$ | 59.8 | 25 | 0.87 |
| High Melting Point Explosive (HMX) | 2691-41-0 | 296.20 | 276–286 | - | $3.33 \times 10^{-14}$ | $2.60 \times 10^{-15}$ | 6.63 | 20 | 0.26:0.06 |
| Nitrobenzene | 98-95-3 | 123.11 | 5.7 | 210.8 | 0.245 | $2.40 \times 10^{-5}$ | 1800 | 25 | 1.85 |

**Table 4.** *Cont.*

| Energetic Compound | CAS Number | MW (g) | Melting Point (°C) | Boiling Point (°C) | Vapor Pressure (mm Hg) | Henry's Law Constant (atm m³ mol⁻¹) | Water Solubility (mg L⁻¹) | Temperature at Solubility Reported (°C) | Log P |
|---|---|---|---|---|---|---|---|---|---|
| Nitroglycerin (NG) | 55-63-0 | 227.09 | 2.8; 13.5 | 218, explodes | $2.00 \times 10^{-4}$ | $4.30 \times 10^{-8}$ | 1800 | 25 | 1.62 |
| Pentaerythritol Tetranitrate (PETN) | 78-11-5 | 316.15 | 140 | 180 | $1.04 \times 10^{-10}$ | $1.20 \times 10^{-11}$ | 43 | 25 | 1.61 |
| Terephthalic Acid | 100-21-0 | 166.13 | 140.6 | 288 | $9.20 \times 10^{-6}$ | $3.88 \times 10^{-3}$ | 15 | 20 | 2 |
| Trinitrophenyl-methylnitramine (Tetryl) | 479-45-8 | 287.15 | 130–132 | 187, explodes | $4.00 \times 10^{-10}$ | $1.00 \times 10^{-11}$ | 75 | 20 | 2.4 |

\* -: Not available.

**Table 5.** Concentrations of common ECs (mg kg⁻¹) in surface and subsurface soils in military-impacted soils in reviewed studies.

| Study | Country | Activity | HMX | RDX | TNT | TNB | 4ADNT | 2ADNT | NG | 2,4-DNT | 2,6-DNT | 1,3,5-TNB | Aminos | Tetryl |
|---|---|---|---|---|---|---|---|---|---|---|---|---|---|---|
| [8] | Korea | MBA * | ND | 51.2 | 53.1 | ND | ND | ND | ND | ND | ND | ND | ND | ND |
| [101] | Korea | MBA | 0.087 | 0.806 | 0.169 | ND | ND | 0.038 | ND | ND | ND | ND | ND | ND |
| [93] | U.S. | MBA * | ND | ND | ND | ND | ND | ND | <0.02–69.64 | <0.014–1.51 | <0.18 | ND | ND | ND |
| | | 22.9–30.5 cm | ND | ND | ND | ND | ND | ND | 0.06–3.35 | <0.014–0.11 | <0.18 | ND | ND | ND |
| | | 45.7–60.9 cm | ND | ND | ND | ND | ND | ND | <0.02–0.69 | <0.014–<0.14 | <0.18 | ND | ND | ND |
| | | 76.2–91.4 cm | ND | ND | ND | ND | ND | ND | <0.02–0.67 | <0.014 | <0.18 | ND | ND | ND |
| [13] | Canada | MBA * | 20–1470 | 1.4–6000 | 40–500,000 | ND | ND | ND | 4–500 | 25–760 | 6.52–270 | 4810 | 5.7–4420 | 3390 |
| | | MBA * (crater soil) | ND | ND | 79,000 | ND | ND | ND | ND | ND | ND | 350 | ND | ND |
| [96] | U.S. | MBA * (demolition ranges) | 0.04–4.63 | 0.06–28.61 | 0.05–234.05 | ND | ND | ND | 0.21–10.74 | 0.08–0.8 | ND | ND | ND | ND |
| [92] | U.S. | MBA * | 600–900 | 800–1900 | 4000–10,000 | ND | ND | ND | ND | ND | ND | ND | ND | ND |
| [6] | Korea | MBA | 0.0165-0.470 | 0.00203–13.4 | 0.00306–0.058 | ND | ND | ND | ND | ND | ND | ND | ND | ND |
| [96] | U.S., Canada | MBA * | <0.01–745 | <0.01–5.1 | <0.01–73 | <0.01–0.15 | <0.01–0.01 | <0.01 | 2.5–3.58 | 0.91 | <0.01 | ND | ND | ND |

**Table 5.** *Cont.*

| Study | Country | Activity | HMX | RDX | TNT | TNB | 4ADNT | 2ADNT | NG | 2,4-DNT | 2,6-DNT | 1,3,5-TNB | Aminos | Tetryl |
|---|---|---|---|---|---|---|---|---|---|---|---|---|---|---|
| [111] | U.S. | MBA * | 0.003–94 | <0.001–825 | 0.005–537 | 0.001–4 | <0.01–0.05 | <0.01–0.11 | 12 | 0.11 | <0.01 | ND | ND | ND |
| [109] | U.S. | MBA * | ND | ND | ND | ND | ND | ND | <0.01 | 0.66–9.1 | 0.35 | ND | ND | ND |
| [97] | U.S. | MBA * | 489–874 | 0.5–1 | 2–6 | ND | 0.4–0.8 | 0.5–0.7 | 35–34 | ND | ND | ND | ND | ND |
| [98] | U.S. | MBA * | 0.46–5.6 | 2.1–6.5 | 0.62–5.6 | <0.01–0.01 | 0.23–0.51 | 0.31–0.61 | ND | ND | ND | ND | ND | ND |
| [102] | U.S., Canada | MBA * | 0.53–9.1 | 5.6–51 | 0.78–36 | <0.01–0.28 | <0.01–0.40 | <0.01–0.03 | 0.35 | 0.04 | <0.01 | ND | ND | ND |
| [107] | Canada | MBA * | <0.01 | <0.01 | <0.01 | <0.01 | <0.01 | <0.01 | ND | ND | ND | ND | ND | ND |
| [91] | Canada | MBA * | 0.05–0.19 | 0.45–0.71 | <0.01–0.06 | <0.01 | <0.01–0.02 | <0.01–0.02 | ND | ND | ND | ND | ND | ND |
| [103] | U.S. | MBA * | 0.02- 302 | <0.003–1130 | <0.001–2520 | <0.01–148 | <0.002–12 | <0.01–18 | <0.001 | 0.97 | <0.001 | ND | ND | ND |
| [106] | U.S. | MBA * | <0.01 | <0.01 | <0.01–42,200 | <0.01 | <0.01 | <0.01 | ND | ND | ND | ND | ND | ND |
| [112] | U.S. | MBA * | <0.03–23 | <0.003–54 | <0.001–9440 | <0.003–50 | 0.01–0.05 | 0.007–0.12 | 1.85–26 | <0.01–3.2 | <0.02–4.6 | ND | ND | ND |
| [95] | U.S. | MBA * | 1.0–1.8 | 4.4–7.5 | 1.5–15,100 | 0.05–15 | 0.15–110 | 0.13–102 | <0.01 | 4.0–84 | <0.01 | ND | ND | ND |
| [108] | U.S. | MBA * | 40 | 340 | 130 | 0.2 | 1 | 0.8 | ND | 0.04 | ND | ND | ND | ND |
| [94] | Canada | MBA * | 0.02 | 0.12 | 0.12 | <0.01 | <0.01 | <0.01 | ND | ND | ND | ND | ND | ND |
| [99] | U.S. | MBA * | 307 | 0.25 | 0.20 | ND | 0.69 | 0.55 | NA | ND | ND | ND | ND | ND |
| [105] | U.S. | MBA * | 399–987 | 0.1–5.3 | 3–126 | ND | ND | ND | 7.8 | ND | ND | ND | ND | ND |

* Surface soil samples, MBA: Military base area.

Based on the reviewed studies and their findings, it could be stated that the fate of explosive compounds is governed by both abiotic and biotic processes after their introduction into the terrestrial environment [4,79,100]. The physiochemical properties of ECs, along with environmental and biological factors, play a significant role in the rate and extent of their transportation and transformation. The transport of explosives is influenced by the main processes of dissolution, volatilization, and adsorption whereas their transformation is affected by the main processes of photolysis, reduction, hydrolysis, and biological degradation [4,87].

In the biosphere, the primary mechanism of transport and dispersion of explosives is dissolution in water [4,87,104]. It is necessary to mention that numerous studies have addressed the dissolution mechanisms of individual explosives and propellant formulations in soil, but their results might have limited applicability for residuals dissolution in soils of war-impacted areas because of their exact formulation (e.g., being mixed with binders, waxes, and stabilizers which can reduce dissolution rate of individual ECs) [4,104,113,114].

Aqueous solubility is one of the properties of compounds governing their behavior in soil. Solid particles of explosives could be released to the local environment over extended periods due to their generally relatively low aqueous solubility (e.g., HMX: 6.63 mg L$^{-1}$, RDX: 59.8 mg L$^{-1}$, and TNT: 130 mg L$^{-1}$; Table 4) [114]. Due to its low aqueous solubility, HMX tends to accumulate in surface soils in compound form and then could migrate through vadose zone towards groundwater. While TNT has higher solubility, RDX and HMX have been shown to penetrate deeper into the soil profile than TNT [4,104], and they have been detected in groundwater below several training fields where TNT has not [79,115]. Between 2, 4-DNT and 2, 6-DNT, studies on soils showed higher dissolution for 2, 4-DNT than for 2, 6-DNT [113,116]. NG has high solubility (1800 mg L$^{-1}$) and thus is particularly mobile in soils [93]. In the case of ammonium perchlorate, because of its relatively high solubility and negligible partitioning in soil, it is not expected to persist in soil. Solid particulate perchlorate rapidly dissolves and is transported when it is in contact with moisture [104].

The volatilization potential of explosives is generally low. At ambient temperatures, most explosives are in the solid phase (Table 4) [79,100] and sublimation is mostly insignificant. Only a few ECs with Henry's law constant above $10^{-5}$ atm m$^3$ mol$^{-1}$ have considerable potential to volatilize from the aqueous phase. The common explosives TNT, RDX, HMX, NG, 2, 4-DNT, and 2, 6-DNT (with Henry's law constant values from $10^{-7}$ to $10^{-15}$ atm m$^3$ mol$^{-1}$) do not readily volatilize in the aqueous phase [79,100]. Therefore, the volatilization of explosives is a negligible pathway for their introduction to the biosphere [4,87].

The sorption potential of explosives also seems low in general. For example, TNT, RDX, HMX, NG, 2, 4-DNT, and 2, 6-DNT have relatively low octanol-to-water partition coefficient (Log P) values (1.6, 0.87, 0.26, 1.62, 1.98, and 2.1; respectively) and thus are expected to be not strongly sorbed by soils, implying potential mobility in the biosphere [100,104,106,110,115,117]. Some studies suggest that TNT may be reversibly sorbed by soil [118–120]. The suggested sorption mechanisms include hydrogen bonding and ion exchange among soil colloids and the nitro functional group. The soil/water partitioning coefficient ($K_d$) for TNT was in the range of 2.7–3.7 L kg$^{-1}$ in surface soils [121] whereas it ranged from 0.04 to 0.27 L kg$^{-1}$ in aquifer material [122]. Generally, the sorption potential of RDX is lower than that of TNT. Its sorption is minimal but nearly irreversible [112]. $K_d$ values for RDX have been reported in the ranges of 0.06–7.3 L kg$^{-1}$, 0.12–2.37 L kg$^{-1}$ [123], and 0.21–0.33 L kg$^{-1}$ [118]. It is presumed that HMX has less sorption potential by soils than TNT [4,79,87]. In general, $K_d$ values for HMX are reported within the range of 1 to 18 L kg$^{-1}$ in surface soils and <1 L kg$^{-1}$ in aquifer materials [79]. $K_d$ values for NG, 2, 4-DNT and 2, 6-DNT in small-arms areas ranged from 0 to 7.3 L kg$^{-1}$, 0.1 to 21.3 L kg$^{-1}$, and 0 to 18.2 L kg$^{-1}$, respectively.

Soil properties such as CEC, OC content, and clay content have direct and significant effects on the values of $K_d$, which in turn affects the retention of the contaminant of concern. OC in soil plays an important role in the sorption of explosive compounds [4,116]. Studies showed that $K_d$ values for RDX, TNT, and 2, 4-DNT were dependent on soil organic content. Based on their organic

carbon-water partition coefficient ($K_{oc}$), 2, 4-DNT (360) and TNT (1600) have strong potential for adsorption compared to RDX [124] (Table 4). RDX was found much less associated with soil OM than TNT [79,125]. Meanwhile, it was determined that HMX adsorption was not significantly affected by soil OM [126].

*4.3. Exposure of Soil to Chemical Warfare Agents*

CWAs are compounds that are used to kill, injure, or incapacitate enemies in wars and military operations based on their toxic properties [11,127,128]. They are classified according to toxicity mechanisms in humans: blister agents, nerve agents, choking agents, asphyxiants, and incapacitating/behavior-altering agents [127]. Toxicological mechanisms of certain CWAs, as well as particular mechanisms of toxic action, have been recently reviewed [129] and thus are not included in the present review. Table 6 shows the physiochemical properties of CWAs [89], each of which carry the potential to pose specific environmental and public health risks as discussed later. CWAs have been widely used/are claimed to be used in many conflicts [130], notable examples being during WWI and Vietnam War.

**Table 6.** Physio-chemical properties of common chemical warfare agents (CWAs) [89,131].

| Name Chemical Formula | CAS Number | Molar Mass (g mol$^{-1}$) | Appearance/Odor | Density (g mL$^{-1}$) | Melting Point (°C) | Boiling Point (°C) | Solubility | Vapor Pressure (mm Hg) | Log P |
|---|---|---|---|---|---|---|---|---|---|
| 2-Chlorovinyldichloroarsine $C_2H_2AsCl_3$ | 541-25-3 | 207.32 | Colorless liquid when pure; impurities lead to colors ranging from violet to brown, faint odor of geranium | 1.89 | −18 | 190 | Greater than 10% in ethanol; greater than 10% in ethyl ether, and in water, 500 mg L$^{-1}$. | 0.58 (25 °C) | - * |
| 2-Diethoxyphosphorylsulfanyl-N,N-diethylethanamine $C_{10}H_{24}NO_3PS$ | 78-53-5 | 269.34 | Colorless liquid | - | 110 deg C at 0.2 mm Hg | 110 °C (230 °F; 383 K) at 0.2 mm Hg | Highly soluble in water and most organic solvents | 0.01 (80 °C) | - |
| 2-(Dimethylamino)ethyl 2,2-diphenyl-2-prop-2-ynoxyacetate $C_{21}H_{23}NO_3$ | 6581-06-2 | 337.40 | Vapors colorless, odorless | - | 164 to 165 | 322 | 200 mg L$^{-1}$ at 25 °C (water), soluble in DMSO, propylene glycol, and other solvents | $2.38 \times 10^{-10}$ (25 °C) | - |
| 2-[Ethoxy(methyl)phosphoryl]sulfanyl-N,N-diethylethanamine $C_9H_{22}NO_2PS$ | 21770-86-5 | 239.32 | - | - | - | - | - | - | - |
| 2-[Fluoro(methyl)phosphoryl]oxypropane $C_4H_{10}FO_2P$ | 107-44-8 | 140.09 | Clear colorless liquid, brownish if impure, Odorless in pure form. Impure sarin can smell like mustard or burned rubber. | 1.0887 (25 °C) | −56 | 158 | Miscible in water | 2.86 (25 °C) | 0.30 |
| 3-[Fluoro(methyl)phosphoryl]oxy-2,2 dimethylbutane $C_7H_{16}FO_2P$ | 96-64-0 | 182.18 | When pure, colorless liquid with odor resembling rotten fruit. With impurities, amber or dark brown, with odor of camphor oil. | 1.022 | −42 | 198 | Moderate in water | 0.40 (25 °C) | - |
| (6aR,9R)-N,N-diethyl-7-methyl-6,6a,8,9-tetrahydro-4H-indolo[4,3-fg]quinoline-9-carboxamide $C_{20}H_{25}N_3O$ | 50-37-3 | 323.44 | Colorless, odorless | - | 80 to 85 | - | 67.02 mg L$^{-1}$ at 25 °C (water) | $2.04 \times 10^{-8}$ (25 °C) | 2.95 |
| Arsine $AsH_3$ | 7784-42-1 | 77.95 | Colorless gas, disagreeable garlic odor | 4.93, gas | −111.2 | −62.5 | 0.07 g 100 mL$^{-1}$ at 25 °C (water) | 11,000 (25 °C) | - |

**Table 6.** *Cont.*

| Name Chemical Formula | CAS Number | Molar Mass (g mol$^{-1}$) | Appearance/Odor | Density (g mL$^{-1}$) | Melting Point (°C) | Boiling Point (°C) | Solubility | Vapor Pressure (mm Hg) | Log P |
|---|---|---|---|---|---|---|---|---|---|
| Bis(2-chloroethyl) sulfide $C_4H_8Cl_2S$ | 69020-37-7 | 159.07 | Colorless if pure. Normally ranges from pale yellow to dark brown. Slight garlic or horseradish type odor | 1.27, liquid | 14.4 | 218 decompose at 217 | Soluble in THF, alcohol, lipids, benzene. | 0.11 (25 °C) | - |
| Carbononitridic chloride CNCl | 506-77-4 | 61.47 | Colorless gas, acrid | 0.003 | −6.55 | 13 | Soluble in ethanol, water, ether | $1.23 \times 10^{+3}$ (25 °C) | - |
| Carbonyl dichloride $COCl_2$, also $CCl_2O$ | 75-44-5 | 98.92 | Colorless gas, suffocating odor like musty hay | 4.248 (15 °C), gas | −118 | 8.3 | Soluble in toluene, benzene, acetic acid but insoluble in water. | 1.216 (20°C) | - |
| Chlorine $Cl_2$ | 7782-50-5 | 70.90 | Pale yellow-green gas, pungent, irritating | 3.2 at STP | −101.5 | −34.04 | 6300 mg L$^{-1}$ at 25 °C (water) | $5.83 \times 10^{+3}$ (25 °C) | - |
| Dichloro(ethyl)arsane $C_2H_5AsCl_2$ | 598-14-1 | 174.89 | Colorless mobile liquid, biting irritant odor | 1.742 (14 °C) | -65 | −156 | Soluble in benzene, alcohol, ether, and water. | 2.29 (21.5 °C) | - |
| Dichloro(methyl)arsane $CH_3AsCl_2$ | 593-89-5 | 160.86 | Colorless liquid, pungent odor | 1.836 | −55 | 133 | Reacts in water | 760 (163 °C) | - |
| Dichloro(phenyl)arsane $C_6H_5AsCl_2$ | 696-28-6 | 222.93 | Colorless gas or liquid | 1.65 (20 °C) | −20 | 252 to 255 | Reacts in water, and soluble in benzene, ether, acetone. | 0.033 (25 °C) | 3.06 |
| [Dimethylamino(ethoxy)phosphoryl] formonitrile $C_5H_{11}N_2O_2P$ | 77-81-6 | 162.13 | Colorless to brown liquid, in small concentrations it smells of fruit but in large concentrations it smells of fish | 1.0887 (25 °C) | −50 | 247.5 | 9.8 g 100 mL$^{-1}$ at 25 °C (water) | 0.07 (25 °C) | - |
| [Fluoro(methyl) phosphoryl] oxycyclohexane $C_7H_{14}FO_2P$ | 329-99-7 | 180.16 | Colorless liquid | 1.1278 | −30 | 239 | Almost insoluble in water | - | - |
| Formonitrile HCN | 74-90-8 | 27.03 | Colorless liquid or gas, oil of bitter almond | 0.6876 | −13.29 | 26 | Miscible in water, and ethanol | 742 (25 °C) | - |
| Mechlorethamine $C_5H_{11}Cl_2N$ | 51-75-2 | 156.05 | Colorless liquid, fishy ammoniacal odor | 1.118 (25 °C) | −60 | 87 at 18 mm Hg | Miscible with carbon disulfide, dimethyl formamide, carbon tetrachloride, and many oils and organic solvents And very slightly soluble in water. | 0.17 (25 °C) | 0.91 |

**Table 6.** *Cont.*

| Name<br>Chemical Formula | CAS Number | Molar Mass<br>(g mol$^{-1}$) | Appearance/Odor | Density<br>(g mL$^{-1}$) | Melting Point<br>(°C) | Boiling Point<br>(°C) | Solubility | Vapor<br>Pressure<br>(mm Hg) | Log P |
|---|---|---|---|---|---|---|---|---|---|
| N-[2-[ethoxy(methyl)phosphoryl]<br>sulfanylethyl]-N-propan-2-ylpropan<br>-2-amine<br>$C_{11}H_{26}NO_2PS$ | 50782-69-9 | 267.37 | Amber-colored liquid,<br>odorless | 1.0083 | −51 | 300 | 30 g L$^{-1}$ at 25 °C (water),<br>Dissolves well in organic<br>solvents | $8.78 \times 10^{-4}$<br>(25 °C) | 2.047 |
| Trichlormethine<br>$C_6H_{12}Cl_3N$ | 555-77-1 | 204.52 | Colorless liquid, fishy<br>ammoniacal odor | 1.24 | −4 to −3.7 | 143 | 160 mg L$^{-1}$ at 25 °C<br>(water),<br>miscible with carbon<br>disulfide, dimethyl<br>formamide, carbon<br>tetrachloride, and many<br>other oils and organic<br>solvents | 0.011 (25 °C) | 1.306 |
| Trichloromethyl carbonochloridate<br>$C_2Cl_4O_2$ | 503-38-8 | 197.82 | Liquid at room<br>temperature, odor similar<br>to Phosgene | 1.65 | −57 | 128 | Insoluble in water | 9.75 at (20 °C) | 1.49 |
| Trichloro(nitro)methane<br>$CCl_3NO_2$ | 76-06-2 | 164.38 | Colorless liquid,<br>irritating odor | 1.692 | −69 | 112 | 0.2% in water | 18 (20 °C) | - |

* -: Not available.

Lewisite ($C_2H_2AsCl_3$) is a blister agent that had been developed during WWI and has still been in use during the Iran-Iraq War [11,132]. Yprite ($C_4H_8Cl_2S$), also known as mustard gas or sulfur mustard, is another well-known blister agent that had been used first time during WWI in Belgium and has been still in use during Iran-Iraq War [64,127]. It was speculated that some blistering agents were used on arid soils of the Persian Gulf region during the Iran-Iraq War or the following Iraqi aggression against Kuwait [133]. For the first time, tabun (GA) and sarin (GB) nerve agents have been used in war zones by Iraq against Iran during the first Persian Gulf War and also against the Kurdish rebels (unreacted sarin and its breakdown products were detected) [8,127,130,134].

Although the use of several CWAs has been banned/has stopped [129], their environmental legacy from their use in the past prevails. Chemical warfare material breakdown products are listed in Table 7. It is possible to find many of these in war-impacted areas as the residue of their parent compounds [131], the exact repartition depending on the contaminant properties as well as environmental parameters. Soman (military designation: GD), having low vapor pressure, is less volatile than sarin but undergoes a similar hydrolysis pathway; soman solubilizes at a slower rate than sarin does [135]. Mustard gas quickly vaporizes in dry, warm environments but stays liquid in damp and cold environments. It is rarely soluble in water. Nitrogen mustards are alkylating agents that are rarely soluble in water [127,136]. Lewisite hydrolyzes in water to form hydrochloric acid. In contact with alkaline solutions, Lewisite can form poisonous trisodium arsenate [137]. Arsine is soluble in water (200 mL L$^{-1}$) as well as in many organic solvents. BZ can persist for three to four weeks in moist air and is extremely persistent in water and soil on most surfaces [11]. The nerve agents have the expected impact of killing some biota in soil, which is still under study [11].

**Table 7.** Breakdown products of CWAs [131].

| Constituent | Breakdown Products |
|:---:|:---|
| Ions | Sulfate, Chloride, Nitrate |
| Anions | Arsenate, Arsenite |
| Acids | Hydrochloric acid, Sulfuric acid, Nitric acid |
| Mustard gas | Thiodiglycol sulfone, 1,4-Oxathiane, 1,4-Dithiane, Thiodiglycol sulfoxide, ThiodiglycoL |
| Nerve agent (VX) | bis (2-Diisopropylaminoethyl) disulfide, bis (2-Diisopropylaminoethyl) sulfide, Diisopropylaminoethanol, Diisopropyl ethyl mercaptoamine, Ethanol, Methylphosphonic acid, S-(2-diisopropylaminoethyl) methylphosphonothioate, Ethyl methylphosphonic acid, Ethyl methylphosphonothioic acid |
| Tabun (GA) | Dimethylphosphoramidate, Dimethylamine, Ethyl *N,N*-dimethylamido phosphoric acid, Dimethylphosphoramide cyanidate, Ethylphosphoral cyanidate, Phosphorocyanidate, Hydrogen cyanide |
| Sarin (GB) | Isopropyl alcohol, Hydrogen fluoride, Methylphosphonic acid, Isopropyl methylphosphonic acid |
| Soman (GD) | Pinacolyl alcohol, Methylphosphonic acid, Pinacolyl methylphosphonic acid |
| Lewisite | Chlorovinylarsonous acid, Chlorovinylarsonic acid |
| Other agent breakdown products | Benzothiozole, *p*–Chlorophenylmethylsulfoxide, *p*–Chlorophenylmethylsulfide, *p*–Chlorophenylmethylsulfone, Dimethyldisulfide, DMMP (Dimethyl methylphosphonate), DIMP (Diisopropyl methylphosphonate) |

Most nerve agents are liquid at standard pressure and temperature, with differing volatilization patterns. GB evaporates at the same rate as water, and 36 times faster than GA. One of the most widely used nerve agents, VX, is the least volatile agent which can persist in the ground for as long as 24 h

because of its higher adsorption than other nerve agents (e.g., GB, GA, and GD) [127,138]. Regarding its sorption, VX has a very low affinity for goethite, moderate affinity for montmorillonite, and high affinity for hydrophobic organics [139]. However, its toxicity decreases sharply once it is adsorbed onto mineral colloids [140]. It can be degraded completely after three to four weeks after occurrence, independently of soil type [11,141]. Some studies on VX show that 90% of the agent decomposes within 15 days.

Most of the degradation products of nerve agents are less toxic than their parent materials, but HN-2, VX, ED, and L can form toxic breakdown products which are more persistent than the original ones [11,142]. Lewisite can persist in soil for long periods of time because of its low water solubility [143]. Sarin is expected to decompose in soil within five days (<90%). Sarin's persistence can increase at low temperatures where it can remain on snow for two−four weeks when deposited as droplets [8,130]. In hot and dry conditions, it migrates through soil in gaseous form, having approximately five times the density of air. Sarin's hydrolysis rate is a function of the temperature and the pH of the soil with non-toxic hydrolysis products. Under average weather conditions, liquid tabun is stable within one−two days and two weeks on natural snow [8,130]. Tabun, under a neutral environment, can be degraded to O-ethyl N, N-dimethylamidophosphoric acid and HCN. O-ethyl N, N-dimethylamidophosphoric acid further undergoes hydrolysis to phosphoric acid, via a much slower reaction. In the soil environment, apart from hydrolysis, tabun can be subject to biodegradation, N-dealkylation, and nitrile hydrolysis [8]. From contaminated soil with tabun, about 16 compounds were isolated mostly with percentages below 1%.

### 4.4. Pollution of Soil by Other Contaminants Related to Military Activities

It should also be noted that other important contaminants are also often found in military-impacted zones, among which oil and oil residues are very common and quite problematic. And yet, these compounds have been excluded from the present review. For instance, a high-impact oil contamination incident occurred during the Gulf War (1990–1991) during which a 770 km stretch of coastline from southern Kuwait to Abu Ali Island was affected [11,144]. Being the largest oil spill in the history of the world, estimated at as much as eight million barrels, this fouled the waters of the Persian Gulf as well as the coastal areas of Iran, Kuwait, and the majority the shoreline of Saudi Arabia. In some parts of the Saudi coast, the sediments were found to contain up to 7% of oil. Not only can many oil compounds, as well as their degradation byproducts, be highly toxic, but many oil-contaminated sites could also contain a significant amount of PTEs [144].

## 5. Site Characterization of Military-Impacted Zones

### 5.1. Profiling Potentially Toxic Elements

The remains of war activities could lead to long term environmental issues because of their persistent accumulation in soil, one of the issues being high and available PTE content [63]. This accumulation can reduce the variability as well as the quantity of soil organisms that have essential roles in the mineralization and humification processes of OM along with the formation of vegetation cover [58]. Furthermore, pollutants can be transferred to the other parts of an ecosystem via runoff (reaching surface or subsurface waters) and migrate downward towards the soil profile [19,68]. Therefore, site characterization, including the investigation of the source and the distribution of various PTEs, is needed most of the time. Regarding the site characterization for the military-impacted areas, the literature mainly focuses on the spatial distribution of PTEs [19,70] and the vertical profiling [7,9,15,26,66,68,145–147] of the impacted areas.

An investigation of the spatial distribution of PTEs used soil samples collected from topsoil (e.g., 0–10 cm) at different distances from the source of contamination such as firing lines or via randomized sampling (i.e., from several random locations). In a Spanish shooting and training range, the spatial distribution of the area was assessed for firing lines [19]. The highest total content of Pb,

followed by Zn, was mainly found in berm soils in its lower part. The concentrations of Zn were higher than that of Pb in the soils closest to firing lines. The high concentrations were related both to the proximity to backstop sites and the frequent passage of shooters, causing a movement and subsequent transfer of soil particles. The high concentrations of Pb were attributed to a large amount of waste (fragmented and warped) ammunition, which remains in the soil mainly broken and damaged. In the northeast of Verdun, France, the spatial characterization was assessed for a larger, war-impacted area where a massive amount of ammunition and shells were stored at the end of WWI [70]. It was found that the level of contamination in the site caused by As, Cu, Pb, and Zn was significant and localized. The concentration gradients of PTEs decreased from the center of the site towards the forest. The contaminated area at the northern low topographic point of the site (outside the burning zone) showed the horizontal transfer of contaminants mainly caused by runoff water [70].

Significant site characterization efforts were invested in understanding the vertical profiling and behavior of the pollutants. They typically involved collecting soil samples from different depths on the contaminated areas and investigating changes in pollution levels in the profile. Almost all studies have reported highly contaminated topsoil profiles (0–10 cm) with rapidly decreasing levels with increasing depth, with the exception of certain PTEs having relatively higher levels of accumulation at increasing depths. More specifically, although the contamination is generally limited to the first half-meter depth from the surface of the soil, the penetration of contamination for each tracer metal shows differing characteristics. Knechtenhofer et al. [15] reported that the topsoil samples from a military shooting range in Switzerland were highly contaminated with Pb, Sb, Cu, and Ni. Their concentration decreased rapidly with depth reaching its background values for Pb (at 60–70 cm), Sb and Cu (at 40 cm), and Ni (at subsoil) [15]. The concentration drop was associated with the soil characteristics of the area, which has a silicate rich background and acidic bedrock. Another vertical profiling study on the PTEs contamination (the backstop of small arms firing ranges in Quebec, Canada) showed that the contamination was limited mostly to a 30–40 cm depth from the surface of the soil [66]. It was also confirmed that concentrations had an increase around 70 cm depth, which may be caused by the vertical migration of PTEs in the soil solution under acid rains followed by in-depth re-precipitation.

Sanderson et al. [68] investigated the vertical profiles of Pb and Sb in four shooting ranges with different soil characteristics in Australia. Typically, Pb was largely retained on the surface soil and showed different levels of migration into the subsoil; 6–18% for the alkaline soil (pH of 9.3) and 7.5–46% for the acidic soils (pH of 5.4–6.4). Sb was more mobile than Pb, especially in alkaline soil, due to increasing negative charge and the adsorption of Sb(V) to hydroxide and the higher solubility of Sb(III) and Sb(V). The vertical leaching of Sb towards subsoil ranged from 13–100%. Tomic et al. [9,10] reported higher Pb concentrations in the deepest layer (60–100 cm) for the samples collected from a destruction zone for ammunition, mines, and explosive devices located in Bosnia and Herzegovina. They attributed this unexpected finding to the natural origin of Pb due to natural lithogenic and pedogeneous processes rather than the anthropogenic pollution. Cd concentrations noticeably dropped through the profile depth; however, Ni concentrations were similar or higher compared to the concentrations in the surface and subsurface soil layers, indicating its relatively high mobility. Zn concentrations followed typical decreasing concentration gradients towards deeper soil layers. Saulius and Greiciute [147] reported higher concentrations of Pb, Zn, and Cu in subsurface layers, however, which they attributed to the course of time and migration of PTEs.

## 5.2. Profiling Energetic Compounds

ECs are among the sources for off-site migration of various compounds in surface waters and groundwater. Although the site characterization for ECs potentially present in the surface and subsurface soils at military-impacted areas needs more attention, the number of studies is limited [95,104,145]. The analysis of the topsoil samples reported in these studies varied in surface concentrations of ECs even in the same sampling spots over different periods due to the variations in climate and activity densities. For example, in the open detonation facility of the Aberdeen Proving

Ground, U.S. [145], RDX concentration had a 64% reduction from June to September, while DNX concentration had a 100% increase over the same time interval. Jenkins et al. [95] and Pennington et al. [104] analyzed soil samples at Fort Lewis training range in the U.S. They reported that the highest concentrations of residues of explosives were detected in the top few centimeters of the soil. They further showed that the explosive concentrations in surface soils were spatially very heterogeneous within this range, even over short distances. Pennington et al. [104] also conducted a site characterization study for the residues of high explosives compounds on training ranges at Fort Leonard Wood, Canada, at an anti-tank rocket range where NG was the primary detected EC, with concentrations much lower between the firing line and the target (from 0 to 25 m behind the firing line).

Vertical profile studies reported disparities in results with uniform or decreasing trends, and some detected levels only in mid-layers, or lower concentrations followed by increasing levels by depth. RDX concentrations in the Aberdeen Proving Ground, U.S., were the lowest between 120 and 180 cm depth, whereas the highest concentrations were encountered in the depth range of 240–300 cm. There were comparable DNX and TNX concentrations throughout the soil depth. HMX was detected with a uniform concentration in all depth intervals. Finally, TNT was detected only once at a depth of 180–240 cm [145]. The ratio of TNT to 2-ADNT and 4-ADNT reported in the Fort Lewis training range, U.S., was generally higher in the surface soil than at the 10-cm depth [95]. It was attributed to wetter soil at depth, creating a favorable condition for biotransformation. Fort Leonard Wood samples [104] had the highest concentrations of 2ADNT and 4ADNT in some subsurface samples because they were formed as dissolved TNT, which moves through the soil. RDX and HMX could penetrate deeper into the soil profile than TNT, due to the susceptibility of TNT to attenuation reactions with soil components and the lower soil/water partition coefficients for RDX and HMX relative to TNT. HMX and RDX were found in groundwater below several training ranges, but TNT was not detected. NG concentrations decreased from the surface to less than 15% at a depth of 20–27 cm, and to about 1% at a depth of 40–60 cm. The highest concentrations of HMX, RDX, and TNT have been detected in the surface soils.

## 6. Risk Assessment of Military-Impacted Sites

The contaminants in soil and water at military-impacted areas could pose significant risks to human health and the environment because of their potential toxic effects [1,148–152]. The assessment of risks largely differs in terms of methodology depending on whether it is performed considering ecological or human health risks. For the same site, the results may also differ, e.g., a meta assessment from China targeting five ranges showed that the contamination by Pb and to a lower extent other PTEs such as Cu, Hg and Sb can cause various potential ecological risks at all the surveyed ranges whereas only Pb at three out of five ranges leads to possible health risks [153]. It should also be noted that most studies in the literature on military-impacted areas have focused on the human health risk assessment; however, there is evidence that ecological risks can also be significant, e.g., as recently suggested by Dvorak et al. [154] for a military training area in the Czech Republic regarding the accumulation of Hg, Cd, and Pb in wild fish.

Humans can be exposed to these soil contaminants via different pathways: direct contact with soil via ingestion, inhalation, and dermal contact [155], or via consumption of food of animal or plant origin from polluted sites [3]. A notable example in terms of military-impacted zones is the case of Agent Orange (contaminated with the dioxin TCDD) widely applied during the Vietnam War; in hotspots of contamination, the concentrations of TCDD in fish and shrimp are still found to be high after 50 years of its application and fishing near contaminated areas remains banned [152]. Particularly, oral exposure has been considered in the literature in general as the key exposure pathway whereas inhalation and dermal pathways previously seen as pathways of secondary importance have started to receive more attention from researchers [156]. That being said, some recent evidence indicates that six PTEs (Pb, Al, As, Cd, Gd, and U) were significantly higher in the residents of an island settlement located 18 km upwind from a bombing range in comparison to the human subjects from the main island in Puerto

Rico, suggesting the importance of the inhalation pathway (in this case, the scenario being exposure to inhalable resuspended particles resulting from bombing) [157]. When focusing on the quantitative evaluation of impacts to potential receptors, the outcome of a risk assessment may serve as scientific and practical evidence in the decision-making step of environmental management. Therefore, a proper execution of risk assessment on pollutants in contaminated areas has become an essential procedure for managing hazardous substances and developing remediation strategies [6,119,158,159]. The toxicity of contaminants via the oral route to mammals, amphibians, birds, and reptilians has also been reported via studies of acute and chronic toxicity of these compounds [1].

Intertwined components of the full risk assessment framework employed in the literature, including the evaluation of military-impacted sites, are summarized in Figure 3. The majority of the reviewed studies from earlier dates include Level 1 activities along with some of the Level 2 activities, e.g., consideration of site-specific soil characterization and selection of reference values (Figure 3); only a few publications utilizing a complete risk characterization and assessment including the activities in Levels 1 to 3 are present [6,101]. Risks to neonatal or reproductive health from PTEs may require partially different approaches than this framework. For example, recent evidence from Gaza, Palestine points out to that the accumulation of PTEs in the human body following exposure whilst residing in attacked buildings in warfare zones may predispose women to negative birth outcomes [160]. Similarly, in Nasiriyah, Iraq, an inverse association was found between distance to Tallil U.S. Air Base and risk of congenital anomalies and hair levels of thorium and U, warranting further investigations to understand the scope of war contamination and its impact on congenital anomalies [161].

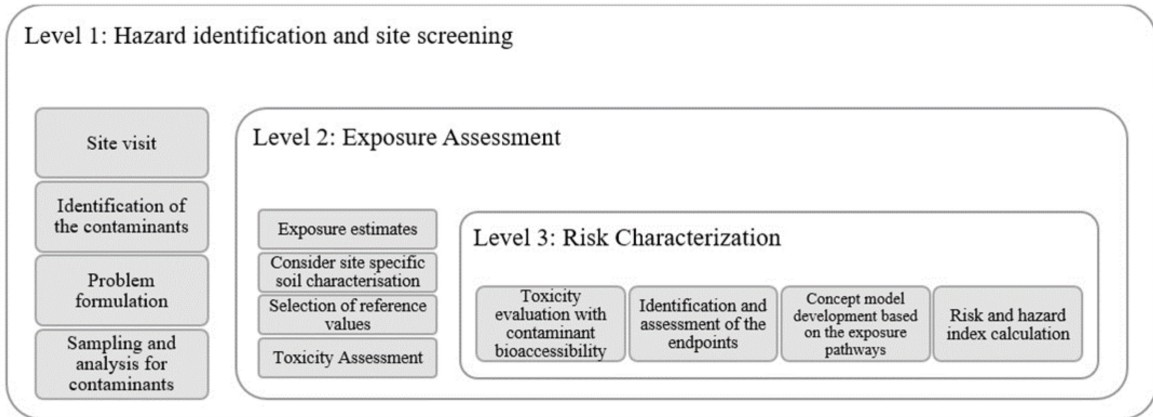

**Figure 3.** Levels of risk assessment method for military-impacted sites (adapted from Jung et al. [101], Ryu et al. [6], and U.S.EPA framework for risk assessment of Superfund sites [162]).

One of the important risk assessment studies for military-impacted sites literature has been performed for a military gunnery range (located in Y-gun of Gyeonggi-do province, Korea) to assess the exposure health risks by the Korean Army for over twenty years [6]. Based on the type of land use and the contents of existing pollutants, a severe bombing zone, as well as a leaching-impacted zone, were separately considered in the assessment. The risk assessment has been performed for three explosives (TNT, RDX, and HMX) and three PTEs (Cd, Cu, and Pb). A site-specific conceptual site model considering reasonable and adequate exposure pathways was constructed to prevent any overestimation of the risk. The American Petroleum Institute's (API) Decision Support System for Exposure and Risk Assessment [163] was used for the calculation of risks. The HI (Hazard Index, used for evaluating non-carcinogenic risks), as well as Risk (carcinogenic risk, expressed as probability) values, were above acceptable thresholds (i.e., HI > 1, Risk > $10^{-4}$ to $10^{-6}$ range). Thus, it was concluded from the risk assessment that there is an immediate need for remediation of both carcinogens and non-carcinogens before construction of a reservoir. Jung et al. [101] assessed human health risks from an open firing range in Korea for civilians during their site visit. TNT, RDX, HMX, and 2-ADNT were

selected as the contaminants for the risk assessment. Korean Exposure Factors Handbook (KEFH) [164] was used to determine the exposure variable values. The default values in the U.S.EPA's guidance [162] were used for those variables that were not listed in the KEFH. Both HI and Risk values were below previously given thresholds, indicating that the site was safe for students and their teachers visiting the site.

Despite the danger, environmental regulations for the potentially toxic substances have not yet been established in many countries which makes the risk assessment of ECs and related PTEs in military zones even more crucial [1,6]. To redevelop the contaminated lands, the U.K. government followed a "suitable for use" approach under Part IIA of the Environmental Protection Act of 1990 [14,165]. The land is considered contaminated when the current or intended use of a site can potentially cause an unacceptable danger to human health or the environment and be assessed for redevelopment on a site-specific basis. In the past, PTE concentrations in the soil were assessed based on total concentrations of a contaminant in the soil, and later the practice changed to their comparison to metal-specific "trigger values" [162]. After the introduction of the Contaminated Land Exposure Assessment (CLEA) [166], these trigger values were replaced by generic soil guidance values (SGVs) [167]. In case of exceeding the SGV, risk assessment or remediation measures are recommended for the studied site [167]. Some further risk management actions are also required when the soil concentrations exceed SGV. To have a better indication of risk associated with a specific site, it is recommended to conduct the risk assessment based on not only SGVs but also by considering commonly measured geochemical and population parameters [21,168].

Finally, to properly assess potential risks from soil contaminants, the knowledge of chemical speciation is necessary in certain cases since elemental, inorganic, and organometallic species of some PTEs have different values of volatility, solubility, reactivity, toxicity, and bioavailability [155,169]. Furthermore, using a reference dose of a contaminant in risk assessment without including its bioavailability may overestimate the risk. As a result, risk characterization studies need to include contaminant bioaccessibility [155] along with taking into account the reasonable and adequate exposure pathways. To conclude, research conducted on human risk assessment of military-impacted areas is still limited [6,101], thus further studies taking these recommendations into account are recommended.

## 7. Conclusions

The present review aims to provide a critical discussion on the environmental effect of military activities (training and warfare) on soils, approaches to contaminated site characterization, and the progress and challenges in human health risk characterization. Soil physical disturbances via excavation of tunnels or trenches, compaction by traffic of troops and machinery, and cratering by bombs (bombturbation) sometimes drastically affect soil such as its topography and hydraulic properties which may lead to landslides and erosion. Chemical disturbances via the introduction of a wide list of potentially toxic elements (PTEs), energetic compounds (ECs), and chemical warfare agents (CWAs) may adversely impact ecosystems as well as human health. Attempts to control these substances are similar to the attempts to control industrial contamination, e.g., via defining and enforcing maximum allowable environmental concentrations. However, the availability of generic limits and legislation is limited to a few countries. Furthermore, these limits do not always include all contaminants of concern, do not overlap in terms of content, and their scientific basis is not always clear. Therefore, a unified and comprehensive scientific framework adequately covering a range of contaminants is needed. There are numerous PTEs found in military impacted zones. Many of these (such as Pb (which is highly toxic, abundant, and persistent), Hg, As, Cd, Cu, and Ni) have been commonly found in elevated concentrations in military-impacted zones. Although their concentrations were extremely high in certain cases, PTE mobility rather than total concentration should be the main concern in site characterization. Although some PTEs naturally tend to be mostly immobile (e.g., Pb), it should be emphasized that changing environmental conditions (e.g., soil pH due to acid rain or oxidoreduction potential due to floods) may increase the mobility of some PTEs. ECs and CWAs seem to

be less common in soils than PTEs but may not be less problematic. For an efficient site characterization, their physiochemical properties (e.g., solubility, Henry's constant, octanol-water partition coefficient) as well as their biodegradability should be first well understood as the soil compartment they will be found in as well as their environmental fate and transport is highly dependent on these factors. The site, then, should be properly characterized based on this information with adequate spatial and vertical profiling. It has been observed that the majority of the site characterization studies included such proper profiling. That being said, contaminant speciation, fractionation, and mobility have not been adequately considered in most studies. The human health risk assessment of military-impacted sites mainly followed a well-agreed framework for evaluating chemical risks. However, this framework is multi-layered, and the extent (depth) to which it was used varied from study to study. Contaminant chemical speciation and bioaccessibility, which directly affect the results for risk characterization, were largely not considered thus should be properly integrated into the assessment process for more accurate results in the future.

**Funding:** This research was funded by Nazarbayev University Research Fund grant number SOE2017004 and the APC was funded by Nazarbayev University Social Policy Grant (SPG) Program.

**Conflicts of Interest:** The authors declare that they have no conflict of interest.

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
