# Peer review of "Soil Contamination in Areas Impacted by Military Activities: A Critical Review"

_sustainability, doi:10.3390/su12219002_

Round 1

Reviewer 1 Report

The article was focused on a more local problem but with the international impact. Strongly recommend the article to be accepted for publication after this minor revisions listed below.

Fig 1 - The quality is very low and the whole Picture was blured. 

 In the text word enriched was used for the case of the contamination of the soil with heavy metals, I proposed to the authors to use word like eleveted levels for Example. 

In table 5 the Partition coefficient is label as log Kow and then in Table 6 as log P, please this is the same property should be coherent. Make it logP, as well I dont read any explanation about the effect of logP. Please cite some articles about the impact of logP.

Reviewer 2 Report

I really like the paper and it is very important that training and conflict sites be looked at to determine the potential for long term environmental damage.

I did notice that the study was focused primarily on Europe, which is fine but I would like to suggest there are more studies in SE Asia that could have been relevant to the storyline. 

Perhaps this studies should be considered. (The suggested studies are shown in attachment)
